# Research on Safety Resilience Evaluation Model of Data Center Physical Infrastructure: An ANP-Based Approach

Xiaer Xiahou [1], Jialong Chen [1], Bangyi Zhao [1], Zixuan Yan [1], Peng Cui [2], Qiming Li [1,*] and Zhou Yu [1]

1   School of Civil Engineering, Southeast University, Nanjing 211189, China
2   School of Civil Engineering, Nanjing Forestry University, Nanjing 210037, China
*   Correspondence: seulqming@163.com

**Abstract:** With the development of the digital economy, the number and scale of data centers are expanding rapidly. Data centers are playing an increasingly important role in social and economic development. However, a short downtime of a data center may result in huge losses. The safety management of data centers' physical infrastructure is of great significance to address this concern. We applied resilience theory to the safety management of data center physical infrastructures. We analyzed the resilience connotation and evaluated the system resilience using the resilience indexes. The data center infrastructure was regarded as a system of systems. Through theoretical analysis, the resilience framework of data center infrastructures was established, which formed the main dimensions of resilience assessment. The Delphi method determined the resilience indices, and the ANP method was adopted to set up the evaluation model. The results revealed the important indexes affecting data center infrastructure system safety resilience. Based on the findings, this paper argues for improving redundancy and adaptability, paying attention to the resilience management of energy flow and thermal flow, and establishing an automatic systematic data management system. These suggested measures would not only effectively make contributions to the data center infrastructure safety management theory but also provide an important reference for construction industry practices.

**Keywords:** data center infrastructure; safety management; resilience; evaluation system





## 1. Introduction

In recent decades, the booming digital economy, big data, and information technology have spawned the development of new infrastructure. Based on information networks, new infrastructure refers to infrastructure driven by technological innovation [1], such as the 5G base station [2]. Unlike traditional infrastructure aiming at connecting physical space, new infrastructure further expands the connection between the digital and physical world [3], which has become the foundation of the digital economy and plays an increasingly important role in the digital transformation and high-quality development of traditional businesses [4]. Globally, new infrastructure has attracted more and more interest, especially in the post-COVID-19 epidemic era, as increasing digital infrastructure investment and speeding up traditional industry digitization have become important for many countries [3].

Digitalization is the basic element of new infrastructure [5], and accordingly, the data center has become an important part of the new infrastructure [6]. According to statistics and prediction, the total amount of global data is expected to grow from 16.1 ZB in 2016 to 163 ZB in 2025. The data explosion and the rapid development of information and communication technologies have created an ever-increasing demand for data centers [7]. In recent years, data centers have been experiencing a steady growth both in number and size. Data centers play an increasingly important role in economic activities, and thus, even very little downtime can lead to a significant loss of revenue [8]. The average cost associated with unplanned data center downtime is USD 8851 per minute [9]. In this case, the safe and smooth operation of data centers has become more and more important [10].

A data center refers to buildings used to store networked computers. The interconnection between these devices forms a network system responsible for providing various Internet and cloud services such as e-commerce, storage backup, video streaming, and high-performance computing [11]. The components of the data center can be divided into IT infrastructures and physical infrastructures. The IT infrastructure system includes architecture, applications, servers, etc. [12]. The function of the physical infrastructure system, which is the main topic of discussion of this paper, is to provide power and appropriate environmental conditions for IT infrastructures [13]. As an important part of the data center, physical infrastructures play an important role in data center security management. The downtime of physical infrastructures, at any level, has a significant impact on the effectiveness of information technology services [9]. However, with the physical infrastructure system becoming more and more complex, the number of air conditioners, power supplies, and cabinets required has also seen massive growth [14], bringing new and serious challenges for safety management. For example, the power losses of the internal power supply system (IPSS) increase with the increasing number of servers, causing a power supply capacity shortage for the devices in the IPSS [15].

Although there are several studies on the safety management of infrastructure in urban construction and industrial systems, few studies have focused on data center infrastructures, which require more attention [16]. Currently, many countries and international industry associations are issuing certification specifications for data center infrastructure, such as *Data Center Site Infrastructure Tier Standard-Topology (2018CN)*, to guide the construction and operation of data center infrastructure, and many evaluations and improvement methods of reliability and availability are taken into consideration. Some researchers also focus on data center infrastructure management (DCIM) systems; for instance, Matko et al. (2019) presented a new intelligent monitoring and event management method for data center physical infrastructures based on multilayer node event processing [9]. The existing efforts mainly focused on certification grading and management from the topological level as a whole. However, the complex infrastructure system management of data centers has uncertainty, and the traditional safety management paradigm emphasizing reliability and availability is unable to meet the needs of safety requirements. Apart from the overall evaluation classification and risk prevention, the evaluation and optimization methods integrating fault resistance and recovery are needed for the data center infrastructure.

Hence, we adopted the concept of resilience into data center infrastructure safety management. The *Resilience Alliance*, a global research organization, defines resilience as the ability of social ecosystems to absorb or resist disturbances to maintain the same structure and function as before [17]. The idea of resilience emphasizes the Safety-II system security research paradigm of automatic resistance and active recovery [18]. At present, resilience theory has been gradually applied to infrastructure research fields [19]. The connotation of resilience is constantly expanding, and its theoretical basis is greatly enriched. Resilience theory, which has become a new research paradigm in the research and practice of safety science, is considered the highest level of security. At present, the qualitative analysis of infrastructure resilience is mainly performed through interviews, questionnaires, and other forms, using the expert scoring method, the analytic hierarchy process, entropy weight method, or other measurement methods to build an infrastructure resilience evaluation index system [20,21]. This paper focuses on the infrastructure system of a single data center. In this paper, the analytic network process (ANP) is used to systematically and comprehensively study the prevention (absorption) before disturbances, resistance during disturbances, and recovery and adaptability after failure from the perspective of resilience. This study aims to offer an in-depth understanding of the safety resilience of data center infrastructure systems, as well as provide a reference for professional practice about disaster and failure preparedness, safety management, and investment decisions.

This paper is organized as follows. Section 2 presents the current status of research on data center infrastructure and resilience. Sections 3–6 discuss the methodology, results and discussion, and conclusions, respectively.

## 2. Literature Review

The data center can be considered as a complex, interacting system of systems [22], consisting of a whole set of complex facilities. As for the composition and division of the infrastructure of the data center, scholars also have different views on this. Paul Townend (2019) divided the data center infrastructure into physical, power, virtual, and business [22]. W.M. Bennaceur (2018) believes the data center encompasses the surrounding power grid, electrical infrastructure, cooling system, server room, and individual servers, down to the CPU [23]. From the summary of existing research, the data center infrastructure can be divided into IT-related equipment (such as servers, storage, and network switches) and physical infrastructure [24] to support and guarantee the operation of IT hardware, and the physical infrastructure discussed in this paper mainly refers to the latter.

For the physical infrastructure, V. Dinesh Reddy (2017) believes power distribution, heating, ventilation, air conditioning, and security management should be included [25]. Fang Fang (2014) proposed that the physical infrastructure of the data center can be divided into space, power supply systems, cooling systems, fire protection systems, cabling systems, and monitoring systems [12]. Outside academia, in industry practice, the Tier classification, as defined by the Uptime Institute, mainly focuses on power and cooling systems. From the current studies, although there are different subdivisions about the physical infrastructure of the data center, there is a consensus on its functions. The basic functions can be divided into three parts: (a) power supply and distribution, which usually correspond to the electrical system, to satisfy the power requirements of servers, other physical infrastructures, and ancillary functions such as lighting [22]; (b) temperature control, which usually refers to cooling systems or thermal systems, to ensure that the water, air, and rack environment are at the appropriate temperature; and (c) management to realize real-time monitoring and ensure safety. Of course, in addition to these three major systems, other components such as the cabling system also play an important role.

To ensure business continuity, data center systems must tolerate different adverse disturbances to mitigate downtime and improve system availability during long-term operations. The disturbance could be a failure within technical aspects such as hardware failures or a natural or man-made disaster such as a storm, fire, power outages, or terrorist acts [14]. Due to the interdependence of data center infrastructure subsystems, some adverse disturbances might be propagated to other dependent components and escalate into a severe system failure, thus damaging business transactions.

Extensive studies have been carried out on the safety management of data centers. Ahmed, Alvarez, et al. (2021b) studied computing resource allocation and reliability [15]. Chen et al. (2017) presented a fault-tolerant DCN solution referring to a switch [26]. Dong and Zhang (2014) proposed a defense system for cloud platforms, cloud service security, and virtual infrastructure security [27]. Graefe (2015) came up with a method of data backup to realize redundant disaster preparedness [28]. In maintaining the security of the data center, compared with computing systems such as rack-level, server layer, and application layer, the physical infrastructure received less attention [29]. How to evaluate the safety performance of physical infrastructure systematically and comprehensively needs further research.

The safety evaluation of the physical infrastructure of data centers mainly includes availability, reliability, fault tolerance, disaster tolerance, disaster recovery, and data monitoring evaluation. Reliability is the probability of a device or system performing its function adequately under specific operating conditions for an intended period [30]. Downtime, uptime, defects per million operations (DPM), failed operations per million, tried operations, mean time between failures (MTBF), and mean time to repair (MTTR) are the main indicators of reliability and availability [31]. Stochastic Petri Nets (SPN) and Reliability Block Diagrams are the main evaluation and modeling methods [32]. Rocha et al. (2020) proved the influence of power architecture and checkpoint mechanism on the application availability and mean time to failure value [33]. Some scholars divide the reliability evaluation of the power system into six parts, namely system monitoring, system protection, surge and

lightning protection, wiring and grounding, preventative maintenance, and system design and availability [34]. In addition, some scholars explored some new indicators of reliability evaluation. Ahmed et al. (2021) proposed the reliability index of the probability of loss of load, which comprehensively considers the probability of common-mode failure of the uninterruptible power supply (UPS) and the load reduction as the power loss increases [35]. DT and HA are top priorities in robust business continuity plans of any enterprise [14]. In addition to academic research, there are various international certification standards for the evaluation and classification of reliability and usability, such as Data Center Site Infrastructure Tier Standard-Topology, Data Center Site Infrastructure Tier Standard-Operational Sustainability, EN 50600 Standards "Information technology—data center facilities and infrastructures", and ANSI/TIA-942—Telecommunications Infrastructure Standard for Data Centers.

However, the extant research is mainly focused on the traditional field of safety and risk management—little attention had been paid to resilience. The existing research on resilience is more scattered in the subsystems of the data center, and there is little comprehensive research. Electrical systems have received more attention. Parise et al. (2020) enhanced the resilience of electronic systems through the management of distributed topology and business continuity systems [36]. Fang and Yu (2014) optimized the cabling system from some important management aspects such as reach, backward compatibility, power, latency, and cost [12]. For thermal systems, in the resistant phase of resilience, Cheung and Wang (2019) calculated the availability of the entire system with different numbers of redundant equipment and distribution headers [37]. In the recovery phase of resilience, emergency and management during outages received research attention [38]. Cho et al. (2019), based on the thermal performance change of the data center during the cooling system downtime, calculated the minimum time required for the backup system to start through the speed and time of the temperature rise [39]. Although researchers have noted the integral management control systems, the extant studies focus on some sub-stages of resilience. For instance, in the resistance and recovery stages, some scholars proposed some fault data point monitoring methods that can effectively improve the fault recall rate and reduce the false detection rate to improve failure resistance and recovery speed [13,40]. To enhance the reaction speed in the resistance and recovery phrases, scholars studied the ability of monitoring tools to collect and integrate data from different types of facilities and equipment [41]. In the absorption stage of resilience, to take precautions against calamity, studies of a predictive maintenance approach applicability to data center sites were conducted [42]. Based on the summary of research status, the existing studies are confined to the optimization of each subsystem or separate and discrete stages belonging to resilience evaluation, such as the resistance stage, absorption stage, and recovery phase.

The limitations of the current research are as follows.

(a) The current mainstream evaluation and certification systems are based on existing specifications and grades, focusing on reliability and availability, minimizing the occurrence of failures. However, little attention has been paid to the absorption, recovery, and adaptation after failure, and the current literature has failed to reach the understanding level of safety resilience.

(b) The calculations about reliability and availability can only obtain a reliability value, but cannot measure the emergency, response, and recovery capabilities while facing failures. They heavily depend on historical data, which cannot be easily obtained.

(c) The present research on resilience is mainly distributed into subsystems, and there is a lack of systematically comprehensive evaluation of the resilience of the overall infrastructure system integrating topology network and the operation and maintenance system.

## 3. Framework and Methods

### 3.1. Data Infrastructure Resilience Framework

According to the literature review and industry consensus, drawing from Walid Mokhtar Bennaceur (2020) [11], this paper assumes that a data center physical infrastructure is composed of heterogeneous resources divided into three main subsystems [41]: (a) an electrical subsystem (including generators, power transformers, uninterruptible power supplies, distribution units, and so on) providing power; (b) a thermal subsystem (including water chillers, pipes, and cooling tower) controlling temperature; and (c) a management and control system monitoring various kinds of data. The data center infrastructure model architecture proposed by Bennaceur and Kloul (2020) is adopted in this study. Energy flows across the data center infrastructure, offering power required to support cooling systems and IT infrastructure. Waste heat generated by networking systems is passed to the thermal system and undergoes heat recycling to space heating [22]. The network subsystem depends on both the electrical subsystem and the cooling subsystem, and the cooling subsystem itself depends on the electrical subsystem to function properly [11].

Since its proposal, resilience has entailed traditional engineering resilience, ecological resilience, and the evolution of resilience [43]. The connotation of resilience can be interpreted from two aspects: the 4Rs of resilience and the external manifestation of resilience. It is considered that robustness, redundancy, resourcefulness, and rapidity are the four important attributes of resilience [44]. The schematic diagram is shown in Figure 1. The resilience of infrastructure systems can be quantified as a combinative ability of absorption, resistance, recovery, and adaptation [45,46]. Absorption is the ability of the system to eliminate potential safety hazards to prevent disasters; resistance capacity is the ability of the system to minimize the impact of disasters to reduce system losses; recovery capacity is the ability to adjust the state in time to ensure normal operation during disasters; adaptation is the ability to optimize the internal structure of the system after experiencing a disaster to improve the ability to respond to unsafe disturbances again. The resiliency framework of the data center's physical infrastructure is shown in Figure 2.

The goal of infrastructure system resilience is to maintain the service continuity of the loads, demonstrating the ability to resist, absorb, recover from, and adapt to failures, disasters, and interruptions.

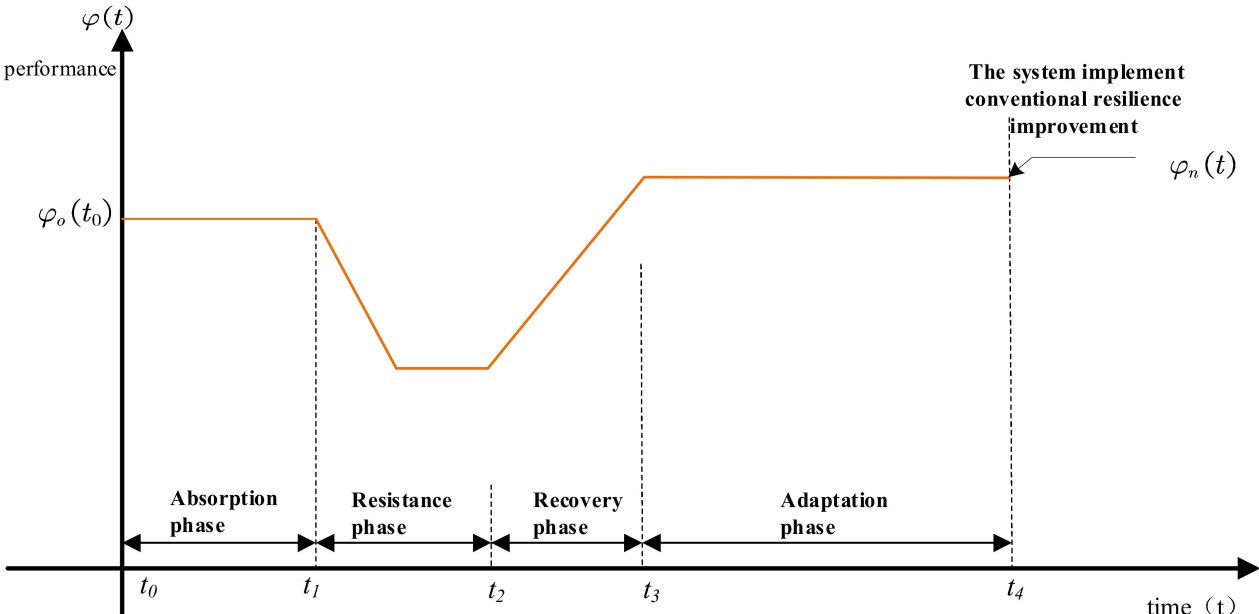

**Figure 1.** Resilience stage diagram.

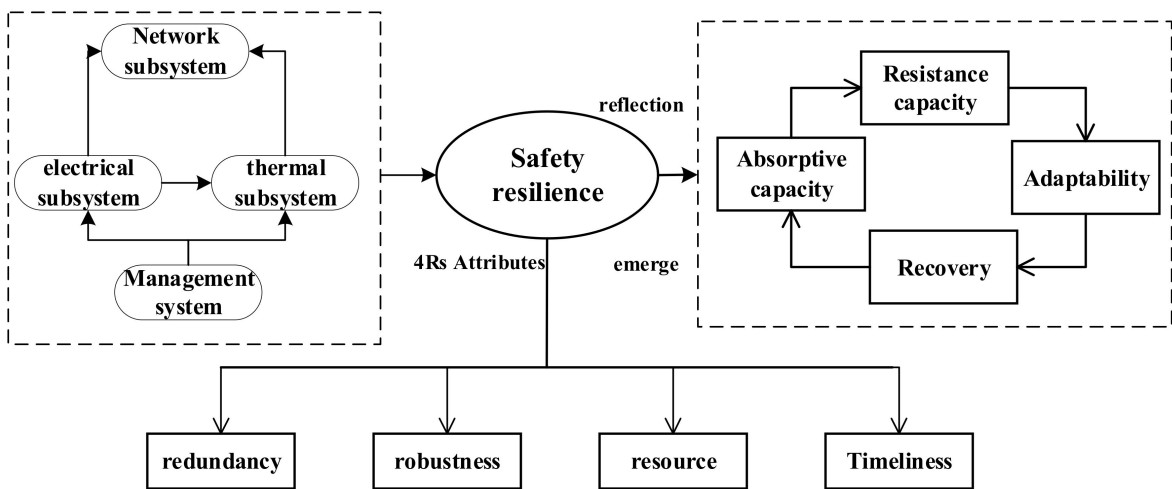

**Figure 2.** Data center physical infrastructure resilience framework.

### 3.2. Selection of Resilience Measurement Indexes and Establishment of Index System

On the premise of ensuring the practicability, hierarchy, and systematicity of the impact indicators of data center infrastructure safety resilience, the index system is constructed, fully considering the 4Rs of resilience, taking the physical infrastructure subsystem of the data center as the indicator dimension, and taking the absorption (Abs), resistance (Res), recovery (Rec), and adaptation (Ada) capabilities as index dimensions. The identification and screening process is shown in Figure 3.

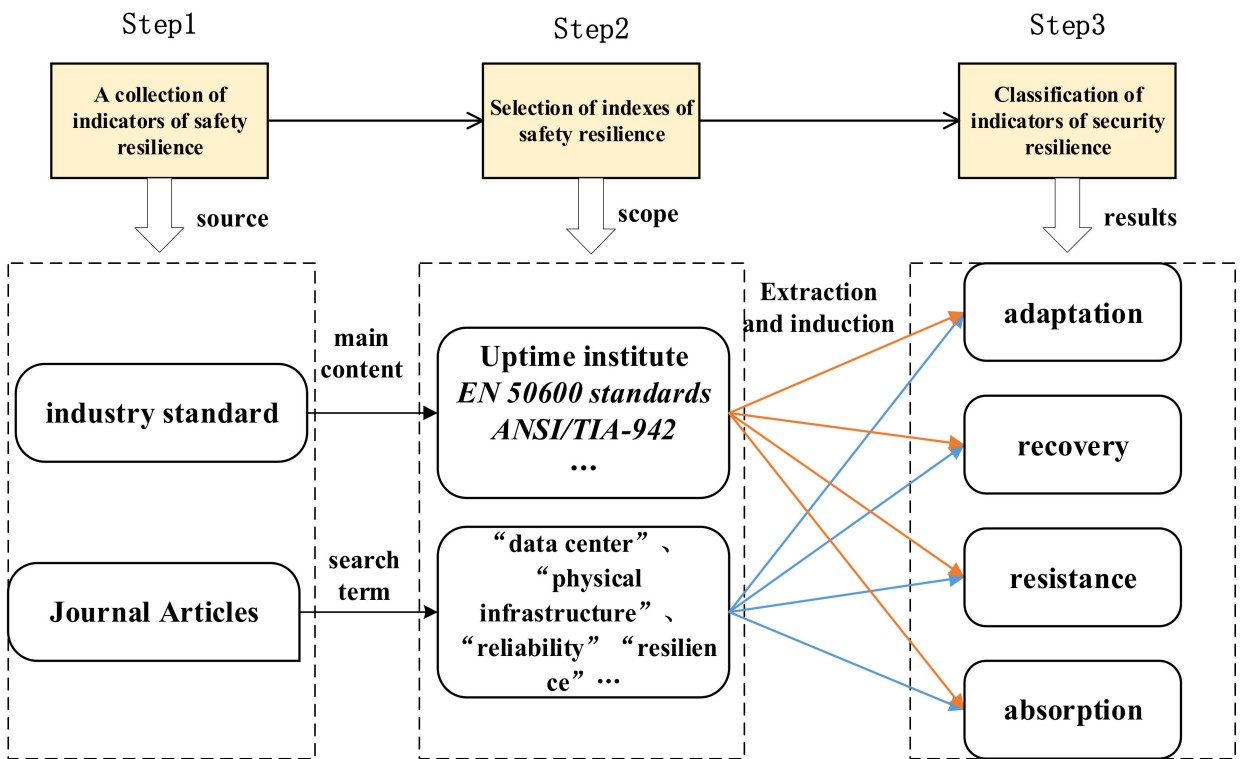

**Figure 3.** Identification process of indicators of data center infrastructure safety resilience.

Firstly, we used the bibliometric method to screen resilience indicators, namely "reliability OR resilience OR recovery OR availability OR failure OR disaster" AND "technical infrastructure* OR physical infrastructure* OR facility" AND "data center", recognized as search terms in the Web of Science database search interface to query. Then, we set the

search year before 2020, and searched a total of 406 WOS core collection papers. Industry Standard Specifications include standard specifications publicly published by the Uptime Institute, EN 50600, and ANSI/TIA-942. The resilience index with a high frequency of statistics was used as the initial index, which is summarized by the ability of absorption, resistance, recovery, and adaptation. According to the connotation and formation mechanism of resilience, 31 indicators were initially screened. The Delphi method was used to screen the indicators. Through expert interviews, the indicators with low common values or serious deviations in the corresponding factor relationships were excluded.

### 3.3. Evaluation Index Model Based on ANP Method

Methods applied to the resilience assessment include the analytic hierarchy process method (AHP) [47,48], system dynamics (SD) modeling [49], the Bayesian network [50], DEMATEL-ISM method [51], and the analytic network process (ANP). SD is mainly used to study the relationship in interdependent complex adaptive systems [52]; however, a large amount of internal data is needed to calibrate various parameters and functions of the model, and it is difficult to model the systemic heterogeneity between infrastructures [53]. Although the Bayesian network effectively calculates the probability to accomplish the evaluation, this method could lead to an incorrect estimation of risk because the relationship of the overlapping factors is not described [54], and the AHP method is weak in describing complex interconnected systems. Compared with AHP and other methods, in the ANP method, the interdependence and influence within groups of criteria in a network structure model can be better reflected [55]. The physical infrastructure of the data center is composed of interrelated infrastructure subsystems, and some internal operation data are difficult to obtain. In this case, the ANP method can accurately and concisely reflect the relationship between components, which is more suitable for the resilience evaluation model of data center physical infrastructure. There are many successful precedents in academia: the ANP method has been widely used in infrastructure resilience assessment such as community earthquake resilience [56], urban infrastructure resilience [57], urban resilience to flooding [58], and interdependent critical infrastructure [59].

We used the ANP method to determine the subjective weights. The analytic network process (ANP) replaces the top–down subordinate relationship of the analytic hierarchy process (AHP) with the interdependence between elements, and replaces the original single hierarchical structure with the feedback network between internal elements, making it more suitable for dealing with scientific decision-making problems of complex systems. The specific steps are as follows.

(a) Establish a structural hierarchy model: Establish a network relationship diagram according to the relationship between the factors of the control layer (objectives and criteria) and the network layer (indicators) by experts. Each relationship is identified by one-way or two-way arrows.

(b) Construction of judgment matrix and determination of local priority: The expert group evaluates the pair-wise comparison matrix of the relationship between various factors according to the structural hierarchy model. After the judgment matrix results are obtained, the consistency test is carried out, and the check coefficient Cr is less than 0.1:

$$CI = \frac{\lambda_{max} - n}{n - 1} \tag{1}$$

$$CR = \frac{CI}{RI} \tag{2}$$

In the formula, $CI$ is the consistency index, $\lambda_{max}$ is the maximum eigenvalue, $n$ is the order of judgment, and $RI$ is the random consistency index.

(c) Calculation of supermatrix: Take control layer element $A_s$ as the control criterion. The weight matrix $F_{ij}$ is constructed based on the sub-criteria of each element in the element group $B_j$. Then, the supermatrix $F_s$ under control criterion $A_s$ is obtained. After

normalization, the weighted supermatrix $\overline{F_S}$ and the convergent limit supermatrix $\overline{F_S}^\infty$ are obtained:

$$F_{ij} = \begin{bmatrix} f_{i1}^{(j1)} & f_{i1}^{(j2)} & \cdots & f_{i1}^{(jn_j)} \\ f_{i2}^{(j1)} & f_{i2}^{(j2)} & \cdots & f_{i2}^{(jn_j)} \\ \vdots & \vdots & \ddots & \vdots \\ f_{in_i}^{(j1)} & f_{in_i}^{(j1)} & \cdots & f_{in_i}^{(jn_j)} \end{bmatrix} \tag{3}$$

$$F_s = \begin{bmatrix} F_{11} & F_{12} & \cdots & F_{1n} \\ F_{21} & F_{22} & \cdots & F_{2n} \\ \vdots & \vdots & \ddots & \vdots \\ F_{n1} & F_{n2} & \cdots & F_{nn} \end{bmatrix} \tag{4}$$

$$\overline{F_S} = \left( \overline{F_{ij}} \right), \overline{F_{ij}} = m_{ij}\left( F_{ij} \right), i = 1, 2, \ldots, n \tag{5}$$

$$\overline{F_S}^\infty = \lim_{t \to \infty} \overline{F_S}^t \tag{6}$$

where $\left[ f_{i1}^{(jl)}, f_{i2}^{(jl)}, \ldots, f_{in_i}^{(jl)} \right]^T$ represents the normalized eigenvector obtained by comparing each element in the element group $B_j$, and $m_{ij}$ represents the elements of the normalized weighting matrix $M_s$ of the factor group judgment matrix under the control criterion $A_s$.

(d) Calculation of group decision making: To ensure the accuracy of the result, the final result is calculated by the geometric average of the results of various experts. After calculating the geometric average value of each index, the final priority value can be obtained.

## 4. Index Identification and Weight Calculation

### 4.1. Setting Up an Index System

Through the literature review, industry norms, and expert opinions, based on the assessment index dimension and framework established in Section 3.1, the index system was set up according to the index identification and screening method introduced in Section 3.2. Finally, 25 indicators of the resilience evaluation system of the data center infrastructure safety system were screened out, as shown in Table 1.

Absorptive capacity is required to be able to solve the disturbance when or before it occurs, to effectively prevent the disturbance from affecting the infrastructure system of the data center. In this part, the stability, standardization, and reliability of the electrical system [34], thermal system, and cabling system [12] are important indicators to prevent faults and disturbances, controlling the energy, temperature, and transmission of this data center. In the electrical system, the stability of electrical power supply comes from the stability of external high-voltage circuit and internal power distribution. However, due to the stable power supply of UPS, the impact of the external power supply of IT equipment can be ignored [60]; therefore, the two electrical power-related indicators mainly focus on the protection system and distribution system. According to *Tier Standard-Operational Sustainability*, normative management of personnel organizations, completeness of daily maintenance management system, and failure risk prediction and early warning were included. Predicting or detecting the disturbance as early as possible and eliminating it in time is the main function of the absorption stage for management and operation. Another important indicator in the management and control subsystem is the scalability management of capacity and load. The demand for technology and data processing is growing; however, the upgrading speed of infrastructure may not match it. Old equipment brings problems such as declining server efficiency and inadequate performance [60], so it is critical to develop a successful long-term design [24], and capacity and load should be flexible and elastic.

**Table 1.** Data center physical infrastructure safety resilience indicator system.

| Resilience Characterization | Subsystem | Specific Manifestations | Extraction of Indicators | Coding |
|---|---|---|---|---|
| Absorptive capacity | Electrical Subsystem | Coordination of ground fault protection, lightning protection grounding, surge current, and protection. | Stability and reliability of electrical system protection | AbsE1 |
| | | UPS input power distribution, UPS output power distribution, load rack row power distribution (pillar cabinet), rack power distribution. | Stability and reliability of electrical distribution system | AbsE2 |
| | Thermal Subsystem | Keeping data center infrastructure temperature stable with water cooling, airflow cooling. | Temperature control stability | AbsT1 |
| | Management and Control Subsystem | Through the tracking information monitoring and management platform, the data are analyzed to identify and predict risks and deal with them. | Failure risk prediction and early warning | AbsM1 |
| | | The process of ensuring that the maximum load is not exceeded to ensure adequate capacity during normal and emergency operation. | Scalability management of capacity and load | AbsM2 |
| | Cabling Subsystem | Reasonable wiring reduces possible unexpected failures, reduces wind resistance, etc. | Specification and stability of cabling system | AbsC1 |
| | Organization and Operation System | Personnel includes the number of full-time personnel, institutions, personnel qualifications, personnel communication mechanisms, etc. | Normative management of personnel organizations | AbsO1 |
| | | Predictive maintenance planning, including delayed maintenance planning, evaluation of repairs, modifications, and redesign of system components. | Completeness of daily maintenance management system | AbsO2 |
| Resistance capacity | Electrical Subsystem | Redundant components ensure switchover and functional assurance in the event of component failure, including backup power system UPS, diesel generators, and energy fuel. | Redundant capacity component of the electrical system | ResE1 |
| | | Features redundant circuit distribution paths that switch in the event of a failure to ensure stable operation. | Multiple independent circuit distribution paths of the electrical system | ResE2 |
| | | Including dynamic power transfer switch and static power transfer switch, which can convert power supply in case of failure, which can be divided into PC-level ATSE and CB-level ATSE. | Power transfer switch | ResE3 |
| | Management and Control Subsystem | Comprehensive and real-time data monitoring; accurate and comprehensive fault detection. | Real-time data and fault monitoring capability | ResM1 |
| | Cabling Subsystem | There is a redundant line emergency in case of line failure, and the compatibility between lines ensures normal conversion. | Redundancy and compatibility of cabling system | ResC1 |
| | Thermal Subsystem | The equipment capacity components of the cooling system have a certain degree of redundancy, such as energy storage devices, coolers, cooling devices, pumps, cooling devices, and fuel tanks. | redundant capacity Components of the thermal system | ResT1 |
| | | The distribution path wiring of the cooling system has some redundancy and can be replaced in the event of a failure. | Multiple independent distribution paths of the thermal system | ResT2 |
| | | Use thermal storage equipment to survive cooler restart times or connect cooling equipment to backup power to maintain adequate backup cooling capacity. | Temperature control and standby energy during downtime | ResT3 |

| Resilience Characterization | Subsystem | Specific Manifestations | Extraction of Indicators | Coding |
|---|---|---|---|---|
| Recovery | Management and Control Subsystem | An efficient process for identifying failure issues with accurate reasoning through data anomalies, identifying root causes, and implementing corrective actions. | Fault monitoring, identification, and reasoning ability | RecM1 |
| | Organization and Operation System | Maintenance list management of installed equipment, special tools, historical data critical spare parts, and reorder points will effectively improve repair performance. | Daily emergency maintenance management system | RecO1 |
| | | Equipped with sufficient and comprehensive emergency management professionals to improve emergency response capabilities through appropriate training to achieve rapid response and effective maintenance. | Organization and reflection ability of emergency personnel | RecO2 |
| | | Quick and effective response through the emergency management system and plan. | Emergency management drill and response | RecO3 |
| | Building Subsystem | Provide sufficient space for maintenance facilities to remove and replace infrastructure equipment safely and quickly. | Effective maintenance space | RecB1 |
| Adaptability | Management and Control Subsystem | Update and upgrade system equipment according to interruptions, and adjust resource allocation capabilities. | System update and resource allocation capability adjustment | AdaM1 |
| | | Record and process the interruption cause and related data to improve the ability to cope with the next interruption. | Historical fault data recording and management | AdaM2 |
| | Organization and Operation System | Accident cause investigation, experience summarization, rectification implementation, and feedback from the operating organization, including records of data, time, root cause analysis, and lessons learned. | Accident summary management system | AdaO1 |

Absorptive capacity refers to the ability of the system to minimize the disturbance impact to reduce the system loss after the disturbance has occurred. In this regard, the main mechanism for the data center to resist disturbance is to set redundancy. In the research and practice of data center infrastructure, redundancy is considered to be a prerequisite for measuring a data center's reliability [60] due to the significant loss of data center outage time. According to *Tier Standard-Topology*, in the electrical and thermal system, the main implementation method of redundancy is to set redundant independent components or allocate paths. To realize the conversion of spare components and paths, power transfer switch [61], temperature control during downtime [38], and real-time monitoring and discovery of faults in management must be taken into account [13].

Recovery requires the system to adjust its state in time to ensure normal operation in a disturbance. At this stage, sufficient building space is necessary to provide maintenance needs. In the management and control subsystem, discovery and reasoning of fault reasons from monitoring data significantly affect the adoption of recovery emergency measures and recovery speed [9]. A daily emergency maintenance management system determines whether the disturbance can be quickly investigated, and the recovery plan can be formulated. When the disturbance develops to the recovery stage, the intervention of operators and emergency management of the operating system are required.

Adaptability is to optimize the internal structure of the system after experiencing disasters to improve the ability to deal with unsafe disturbances again. In this stage, historical fault data recording and management and accident summary management are

the basis for improving the ability to cope with the next disturbance, and can strongly enhance the system capability adjustment ability to reduce the next disturbance risk.

### 4.2. Element Influence Relationship and Weight Calculation Process

Learning from the existing research practices of scholars, taking reference from the existing mature research methods [54,57], the Delphi method and pair-wise comparison method are used to determine the influence relationship and weight of elements. A total of 10 experts (three from design institutes, five from data center operation and management departments, and two from universities and research institutes) were invited to score the data required by ANP. The criteria for expert selection were that they should have 10 years of working experience and have conducted practice or research relevant to the safety management of data centers.

Firstly, we determined the influencing relationship of elements through a questionnaire survey. The materials were distributed to all experts, and the research background, research purpose, research content, and the meaning of relevant indicators were explained in detail. Experts needed to judge the influence relationship between network layer elements independently. After obtaining these data, the correlation of the resilience indicators of the physical infrastructure of the data center was formed. From the statistics of the impact correlation between the network layer indicators, the correlation between the control layer indicators could also be obtained.

We determined the index weight by combining the pair-wise comparison method and the Delphi method. The questionnaire was constructed by the pair-wise comparison method, and the 1–9 scale method was used to judge the dominance between the two factors. The design example of the questionnaire scoring interface is shown in Table 2, and the scoring rules of the 1–9 scale method are shown in Table 3.

**Table 2.** Design of questionnaire scoring interface.

| Reference Index | i | j | k |
|:---:|:---:|:---:|:---:|
| i | | - | - |
| j | | | - |
| k | | | |

**Table 3.** The 1–9 scale method rules.

| Serial Number | Grade of Importance | Cij Valuation |
|:---:|:---:|:---:|
| 1 | i is equally as important as j | 1 |
| 2 | i is equally to moderately more important than j | 2 |
| 3 | i is moderately more important than j | 3 |
| 4 | i is moderately to strongly more important than j | 4 |
| 5 | i is strongly more important than j | 5 |
| 6 | i is strongly to very strongly more important than j | 6 |
| 7 | i is very strongly more important than j | 7 |
| 8 | i is very strongly to extremely more important than j | 8 |
| 9 | i is extremely more important than j | 9 |

In the process of calculating the weight of risk factors, the core link is the solution of the supermatrix, including the solution of the unweighted supermatrix, the solution of the weighted supermatrix, the weight calculation of control layer indicators, the local weight of network layer indicators, and the global weight calculation of network layer indicators. The principle of calculation is given in Section 3.3. Because the whole calculation process is complex, we realized it through *Super Decision* software. The pair-wise importance scores of each index in the questionnaire were input into *Super Decision* software, which quickly constructed the supermatrix and calculated it.

## 5. Results and Discussion

### 5.1. Results

The causal relationship between the indicators in the ANP evaluation model is shown in Figure 4. In the figure, the four large boxes represent the four dimensions of resilience evaluation indicators; each of these boxes contains control-level indicators represented as a small box belonging to each dimension. The one-way arrow indicates one-way influence. For example, the operation energy of the thermal system comes from the electrical system, so the one-way arrow from AbsE2 to AbsT1 means that temperature control stability is affected by the stability and reliability of electrical system protection. Two-way arrows represent mutual influence. For example, daily data recording and maintenance of equipment can enhance understanding of equipment data characteristics, improving efficiency in fault diagnosis reasoning. On the other hand, the enhancement of the reasoning and identification ability of faults can guide the dynamic adjustment of the daily maintenance plan of data center facilities and equipment, so the two-way arrows between RecM1 and RecO1 represent the interaction between the two indicators. The influence relationship between control-level indicators determines the influence relationship between the large-dimensional indicators. For instance, recording and managing historical fault data (AdaM2) can strengthen the data knowledge accumulation of data center electrical and temperature control equipment, influencing the improvement of fault risk prediction and early warning ability (AbsM1). Because the two indicators belong to different large dimension systems, there is an arrow from adaptability to absorptive capacity.

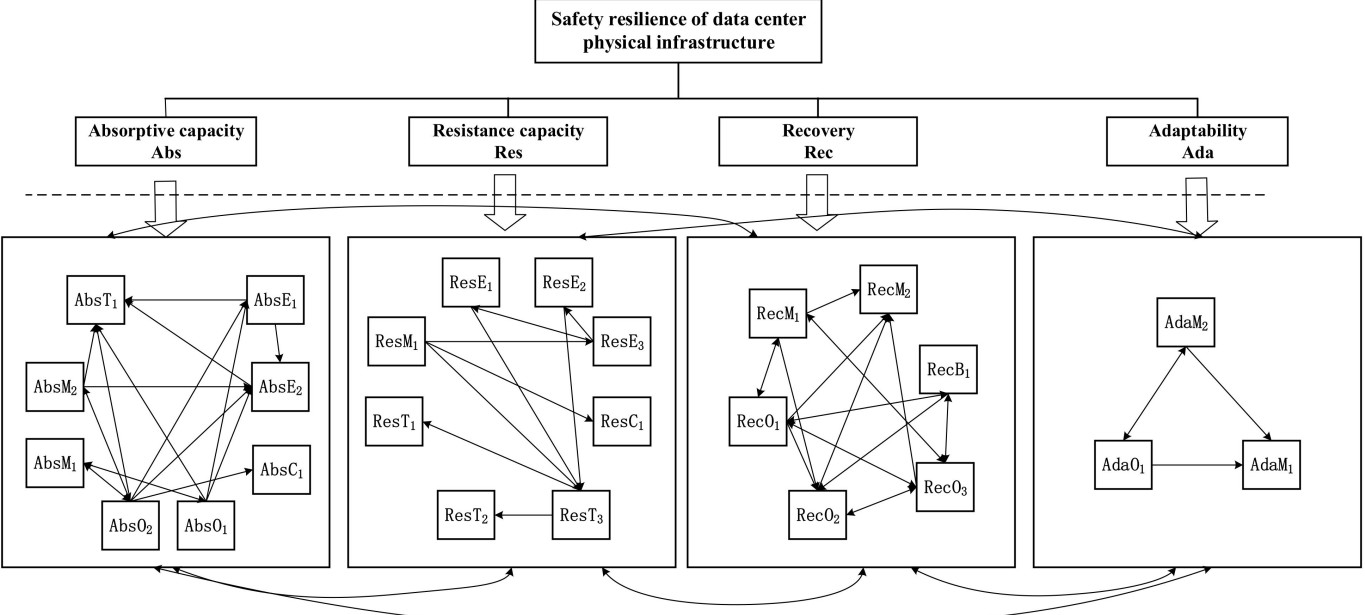

**Figure 4.** ANP causal relationship map.

Due to the complicated calculation process, Super Decision analysis software is used to calculate the weights. The results are presented in Table 4. The fourth column is the weight calculation result of the indexes, the fifth column is the weight ranking of this indicator in all 25 indicators, and the last column is the weight score of the large dimension of resilience.

**Table 4.** Data center physical infrastructure safety resilience index system weight table.

| Cluster | Name | Extraction of Influencing Factors | Limiting | Order | Total Weight |
|---|---|---|---|---|---|
| Absorptive capacity | AbsC1 | Standardization and stability of cabling system | 0.0000284 | 24 | 0.1306677 |
| | AbsE1 | Stability and reliability of electrical system protection | 0.0009553 | 22 | |
| | AbsE2 | Stability and reliability of electrical distribution system | 0.0451186 | 12 | |
| | AbsM1 | Failure risk prediction and early warning | 0.0235201 | 16 | |
| | AbsM2 | Scalability management of capacity and load | 0.0073653 | 19 | |
| | AbsO1 | Normative management of personnel organizations | 0.0004921 | 23 | |
| | AbsO2 | Completeness of daily maintenance management system | 0.0068701 | 20 | |
| | AbsT1 | Temperature control stability | 0.0463178 | 11 | |
| Resistance capacity | ResC1 | Redundancy and compatibility of cabling system | 0.0061931 | 21 | 0.4596134 |
| | ResE1 | Redundant capacity component of the electrical system | 0.0554016 | 9 | |
| | ResE2 | Multiple independent circuit distribution paths of the electrical system | 0.0658952 | 7 | |
| | ResE3 | Power transfer switch | 0.0303343 | 14 | |
| | ResM1 | Real-time data and fault monitoring capability | 0.069567 | 4 | |
| | ResT1 | Redundant capacity components of the thermal system | 0.0612428 | 8 | |
| | ResT2 | Multiple independent distribution paths of the thermal system | 0.0721339 | 3 | |
| | ResT3 | Temperature control and standby energy during downtime | 0.0988455 | 2 | |
| Recovery | RecB1 | Effective maintenance space | 0.0247241 | 15 | 0.1682657 |
| | RecM1 | Fault monitoring, identification, and reasoning ability | 0.0377209 | 13 | |
| | RecO1 | Daily emergency maintenance management system | 0.0191796 | 17 | |
| | RecO2 | Organization and reflection ability of emergency personnel | 0.0186977 | 18 | |
| | RecO3 | Emergency management drill and response | 0.0679434 | 5 | |
| Adaptability | AdaM1 | System update and resource allocation capability adjustment | 0.1188934 | 1 | 0.2414529 |
| | AdaM2 | Historical fault data recording and management | 0.0548175 | 10 | |
| | AdaO1 | Accident summary management system | 0.067742 | 6 | |

### 5.2. Discussion

According to the index model results obtained from ANP, from the weight of four toughness dimensions, the comprehensive weight of resistance capacity is the highest among the four dimensions of resilience, reaching 0.4596134, followed by adaptability, with a weight of 0.2414529, and the last two indicators are absorption and recovery. The index weight ranking is different from that of general construction projects. The safety resilience evaluation of general construction projects tends to show higher weight in the dimensions of adaptability and recovery. The reason behind this is the particularity of redundancy for data center infrastructure. One of the most important design decisions in any data center is the level of redundancy for its infrastructure systems [24]. In case of disturbance, redundant components and paths need to function quickly, and the standby electrical or thermal components must be quickly switched to achieve resistance due to the high loss caused by downtime. The second is adaptability, and it is worth mentioning that the average weight ranking of the three indicators of adaptability is relatively high, which reflects the differences between resilience evaluation and traditional safety evaluation. Resilience emphasizes embracing failure and improving the system's ability to resist disturbance from failure. After an emergency of fault disturbance, the data recording, summary, and feedback system can effectively strengthen the fault identification, reasoning, and maintenance scheme formation of the management control system, improving the stability of risk prediction in advance at the absorption stage and quickly identifying specific failure disturbances to optimize the recovery efficiency in the recovery phase. Therefore, in resilience safety management, under the constraint of limited cost, redundancy and adaptability are the key objects to be improved.

In terms of control-level indicators, the five indicators that have the greatest impact on the safety resilience of the data center infrastructure are system update and resource allocation capability adjustment, temperature control and standby energy during downtime, multiple independent distribution paths of the thermal system, real-time data and fault monitoring capability, and emergency management drill and response. System update and resource allocation capability adjustment rank first in weight. The reasons behind this may be that, on the one hand, the regulation elasticity of capacity and load can enhance the ability of the electrical system and thermal system to cope with fluctuating demand and strengthen the resource setting reserve of system redundancy and recovery. Thus, there are many factors affected by this indicator that make the weight so high. On the other hand, elastic capacity management itself is an important embodiment of resilience. Among the top five indicators, three indicators come from redundancy, one from adaptability, and another from resilience. The indexes under the absorptive capacity system failed to enter the top five, which indicates that compared with traditional safety management, safety resilience argues for the admission of uncertainty and places more emphasis on disturbance resistance and recovery rather than on prevention and stability. The main way to combat fault disturbance is to set redundant paths or allocate resources through the management control system. High redundancy and applicability also reflect the significant difference between data center infrastructure and other infrastructure.

From the perspective of physical infrastructure subsystems, the index weights of the thermal system and electrical system are higher than those of the cabling system and building system. With large capacity and high density, flexibility and scalability cabling are important to ensure the stable operation of the data center. The two important material flows between data center infrastructures, energy flow and heat flow (including water and air), mainly rely on electrical infrastructure and thermal infrastructure. Besides ensuring the stable redundancy of components, it is also necessary to ensure the redundancy of distribution paths. An interesting result is that the average index weight of the thermal system is higher than that of the electrical system. In theory, the normal operation of the thermal system depends on the energy supply of the electrical system. However, the power supply for the data center is relatively stable, but in the complex application scenarios of industry, manufacturing, civil engineering, and rail transit, humid and hot environments often pose greater challenges to data center security management, and this needs our attention.

In terms of overall management control and operation organization indicators, the weights of indicators of the data center management control system—such as failure risk prediction and early warning ability, failure monitoring and identification reasoning ability, system update and resource allocation ability adjustment, and historical failure data recording management—are significantly higher than those of normative management of personnel organizations and emergency personnel organization reflection ability. It is different from general metro and other engineering projects. This is mainly because the management of the data center is becoming more and more digital and automated. Compared with personnel organization management, systematic control is more in line with the large and complex infrastructure equipment of the data center and can more effectively determine the fault and repair and recovery. Among the indexes, real-time data and fault monitoring capability (ResM1) has a high weight of 0.069567, because it directly affects the discovery of fault disturbance. The index of historical fault data recording and management itself hardly directly determines the embodiment of resilience, but it still has a high weight. The reason behind this is that these data affect the representation of AbsM1, ResM1, and RecM1. In DCIM, each infrastructure system is comprehensively managed through data, and the four basic resilience capability dimensions are comprehensively related through the data flow. Therefore, when it comes to the improvement of resilience performance, it should be comprehensively improved from the whole data flow, rather than improving a certain index.

*5.3. Implication*

Based on the research results and discussion, the implications of this paper are as follows.

Data center practitioners need to pay attention to redundancy and adaptability. Unlike other infrastructures and traditional security management, redundancy and adaptability play important roles in the resilience management of data center infrastructure. Therefore, especially in the design stage, the most effective way to improve the resilience performance is to improve the redundancy of electrical, cooling, and other subsystems. Redundancy can be optimized by comprehensively considering the cost according to the safety requirements.

Data center practitioners need to strengthen the resilience construction of energy flow and thermal flow paths. Physical infrastructures in data centers are interconnected and complex, but infrastructures related to energy and thermal flows play a key role in resilience performance. From the findings, we suggest that ensuring the operation and resource supply of the electrical system and thermal system largely determines the safety of the unitary system.

Industry practitioners could build an integrated data management system. There are many infrastructure facilities in the data center. The main way to carry out overall association management is through the infrastructure management system. The importance of the management system in data center resilience is further improved. Resilience indicators for management and operation systems span four resilience phases, and different indicators are interlinked through the flow of management data. It is suggested to establish an integrated data system, break the category gap between various infrastructures, and manage historical data records, daily maintenance, and early warning, discovery, and identification of fault disturbances, to systematically improve resilience.

## 6. Conclusions

With the substantial increase in both the scale and number of data centers, the safety management of data center infrastructures is becoming more and more important. As a system of systems, the safety management of data centers is quite complex, and a comprehensive security evaluation is absent. Therefore, we adopted resilience theory to develop an ANP-based approach for the safety management of data centers.

Through theoretical analysis, the resilience framework of data center infrastructure was constructed, which formed the basic dimensions of the evaluation index system. The index system was determined based on academic literature, industry norms, and expert opinions. The ANP method was adopted to establish the evaluation model. Through weight ranking, some important resilience indicators were found and analyzed. It was found that compared with other infrastructures, the safety resilience evaluation of data center infrastructure has its uniqueness, based on which some targeted suggestions were put forward to improve the resilience management level of the physical infrastructure of the data center.

This paper introduced the resilience evaluation model of data center infrastructure based on ANP. By introducing resilience theory into the research of data center infrastructures, it expands the scope of application of the resilience theory and provides researchers with a new perspective, contributing to producing new research insights. In addition, this paper also considered the physical infrastructure of the data center as an interactive system of systems to perform safety management research, which can enhance scholars' comprehensive understanding of the data center infrastructures. In industry practice, this paper proposes to enhance the redundancy and adaptability, pay attention to the resilience management of energy flow and thermal flow in a data center, and build a systematic data management system. The findings can be adopted as certain reference and decision support for industry practitioners, which would strongly improve the ability of fault disturbance resistance and emergency response.

The definition of disturbance in this paper is extensive, which makes our research universal, but it also brings some limitations. In subsequent studies, the types of disturbances can be refined and more targeted resilience studies can be conducted according to

different types of disturbances. In further research, based on the basic resilience indicator framework of data center physical infrastructure, a more thorough quantitative analysis of system resilience performance and studies to optimize the method of resilience can be carried out.

**Author Contributions:** Conceptualization, X.X., P.C. and Q.L.; methodology, Z.Y. (Zixuan Yan); validation, J.C. and Z.Y. (Zhou Yu); formal analysis, B.Z.; investigation, Z.Y. (Zhou Yu), B.Z. and J.C.; resources, P.C. and Z.Y. (Zhou Yu); data curation, Z.Y. (Zhou Yu), B.Z. and J.C.; writing—original draft preparation, J.C.; writing—review and editing, X.X.; visualization, J.C.; supervision, X.X. and Q.L.; project administration, Q.L.; funding acquisition, X.X., P.C. and Q.L. All authors have read and agreed to the published version of the manuscript.

**Funding:** This research was funded by the National Natural Science Foundation of China (No. 72101054, and 51978164), Ministry of Education in the Humanities and Social Sciences of China (No. 20YJCZH182). The authors also express their gratitude to the experts who participated in this research survey.

**Data Availability Statement:** The data used to support the findings of this study are available from the corresponding author upon request.

**Conflicts of Interest:** The authors declare no conflict of interest.

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
