# Peer review of "Research on Safety Resilience Evaluation Model of Data Center Physical Infrastructure: An ANP-Based Approach"

_buildings, doi:10.3390/buildings12111911_

Round 1

Reviewer 1 Report

The authors developed a safety resilience framework for data centers using the analytic network process method. The paper is well written in general. The reviewer has the following comments and suggestions for the authors to consider:

1. Currently, Section 3 is called “Materials and Methods”. Section 4 is “Index Identification and Weight Calculation”. Section 5 is “Results and Discussion”. It seems that Section 4 is a mixture of methods and results. I would suggest moving some parts of Section 4, which is related to methodology, to Section 3, and some parts of Section 4, which is related to results, to Section 5, and then deleting Section 4. The authors mentioned at the end of the Introduction section that “Sections 3-6 discuss the methodology, results and discussion, and conclusions, respectively”. This statement should correctly reflect the structure of the paper.

Some other minor comments:

1. Table 2 seems to be incomplete.

2. The title of the paper can be improved. Maybe call it “ANP-based safety resilience evaluation model of data center physical infrastructure”?

3. Some sentences are really long. For example, in the abstract, the last five lines are one sentence, which might be hard to read. Please read through the paper and avoid such long sentences (one sentence more than three lines could be too long).

4. A thorough proofreading should be conducted. Some examples: in the abstract, “the safety management level of…” should be “the safety management of”. The last paragraph of the Introduction seems to have a format issue. In section 3.3, “urban Infrastructure resilience [58]” should be “urban infrastructure resilience [58]”.

Author Response

Great thanks for your kind suggestions, we have revised accordingly in the revised manuscripts.

comments 1/  After a careful consideration, we decide to change the head of Section 3 as Framework and method. we still want to keep Section 4 as it was, since it could be considered as the process to conduct the method. We wish you could understand and allow us to do this.

comments 2/ Table 2 is not complete, it is just an example. and we are so sorry to make you misunderstand this problem, and thereby, we made some improvement in Table 2.

comments 3/  the title has also be modified as your suggestion, the new title is Research on safety resilience evaluation model of data center physical infrastructure: an ANP-Based Approch

comments 4 and 5 / sentences and words are checked and we would continue to modify it with the help of the editorial office.

Reviewer 2 Report

Thank you for an interesting paper on an issue that will become increasingly important in the coming years. I was pleased to read a clearly and systematically described methodology for the paper.

I have some minor queries. Firstly, please explain in some more detail how the experts for the Delphi study were selected. Secondly, although you have mentioned the possibility of terrorism I did not see consideration of the physical building where the Data Center is located as opposed to services in that building. I have visited an offsite construction company in my home country where they tested the robustness of their physical building system by using a trial where the armed forces were asked to attempt to break in to the building using all physical force at their disposal. I understand that this is outside your study design. I point it out because physical sabotage by hostile actors is possibly a weak point in any Data Center system. This is just mentioned for your consideration.

I enjoyed reading your paper. May I point out that there is some inconsistency in the use of capital letters?  The first sentence in the Introduction has a capital letter in mid sentence for the word "The". This is not standard English. In addition, sub headigs should have Capital letters and these are missing eg. 5.1 results, 5.2 discussion etc.

Author Response

Thanks for your suggestion,  the following is our response for your kind comments.

1/ The criteria for experts selection are that they should have 10 years working experience and have conducted practice or research relevant to safety management of data center. As in the revised manuscript.

2/ Thanks for your suggestion, we acknowledge that Possibility of terrorism is really out of our research ability, which may inspire our future research. In this manuscript, the possibility of terrorism may embody in factors of ANP models.

3/  The typos have been corrected as your suggestions, and we would continue to improve it with the help of editorial office.

   Thanks so much.

Reviewer 3 Report

The manuscript is well written and accepted in current form. Please revise the abstract. Abstract must contain problem statement ane brief conclusion.

In conclusion section, please rewrite it in the meaningful way rather to talk on other sides. And also please go through the manuscript to correct some grammar mistakes

Best regards 

Author Response

Thanks for your approval, we have modified the abstract and conclusions, and also checked the grammar mistakes and language issues.

We would continue to improve during the processes. 

 Thanks!